# The Role of Death-Associated Protein Kinase-1 in Cell Homeostasis-Related Processes

**DOI:** 10.3390/genes14061274

**Published:** 2023-06-16

**Authors:** Lilian Makgoo, Salerwe Mosebi, Zukile Mbita

**Affiliations:** 1Department of Biochemistry, Microbiology and Biotechnology, University of Limpopo, Private Bag X1106, Pietersburg 0727, Sovenga, South Africa; makgoolilian@gmail.com; 2Department of Life and Consumer Sciences, University of South Africa, Private Bag X6, Johanessburg 1710, Florida, South Africa; mosebs@unisa.ac.za

**Keywords:** apoptosis, cancer, autophagy, cell cycle, DAPK-1

## Abstract

Tremendous amount of financial resources and manpower have been invested to understand the function of numerous genes that are deregulated during the carcinogenesis process, which can be targeted for anticancer therapeutic interventions. *Death-associated protein kinase 1* (*DAPK-1*) is one of the genes that have shown potential as biomarkers for cancer treatment. It is a member of the kinase family, which also includes Death-associated protein kinase 2 (DAPK-2), Death-associated protein kinase 3 (DAPK-3), Death-associated protein kinase-related apoptosis-inducing kinase 1 (DRAK-1) and Death-associated protein kinase-related apoptosis-inducing kinase 2 (DRAK-2). *DAPK-1* is a tumour-suppressor gene that is hypermethylated in most human cancers. Additionally, DAPK-1 regulates a number of cellular processes, including apoptosis, autophagy and the cell cycle. The molecular basis by which DAPK-1 induces these cell homeostasis-related processes for cancer prevention is less understood; hence, they need to be investigated. The purpose of this review is to discuss the current understanding of the mechanisms of DAPK-1 in cell homeostasis-related processes, especially apoptosis, autophagy and the cell cycle. It also explores how the expression of DAPK-1 affects carcinogenesis. Since deregulation of DAPK-1 is implicated in the pathogenesis of cancer, altering DAPK-1 expression or activity may be a promising therapeutic strategy against cancer.

## 1. Introduction

Cancer, a non-discriminatory disease, remains a burden in many countries, including South Africa. Sung et al. [1] estimated that cancer contributed to 10 million deaths in 2020, making it one of the world’s leading causes of death. Additionally, a total of 9.6 million people died of cancer in 2018, of whom 70% lived in low- or middle-income countries [2]. Cancer detection and treatment continue to be every country’s nightmare due to the ineffective nature of the current therapeutic strategies, lack of sensitivity of the current diagnostic approaches, late diagnosis and chemotherapy drug resistance [3,4].

Cancer is a group of malignant diseases that are characterized by abnormal cell proliferation, apoptosis resistance and increased survival signals [5,6]. The loss of proper regulation of the homeostasis processes is often attributed to genetic abnormalities found in cancer cells, and these include mutations in the genes that control cell homeostasis-related processes, including cell proliferation, differentiation, apoptosis and cell survival signalling pathways [7,8]. Cancer and neurodegenerative diseases have been associated with abnormal apoptosis, autophagy and cell cycle deregulation, as well as loss- and gain-of-function of *DAPK-1*. Therefore, it will be interesting to understand how the cell death function of DAPK-1 contributes to its role in cancer.

Cell death by apoptosis is an evolutionarily conserved process that maintains a homeostatic balance between the formation and death of cells [9,10]. Cancer impairs the ability of cells to undergo apoptosis-induced death, allowing them to proliferate uncontrollably. Overexpression of anti-apoptotic proteins is one mechanism by which cancer cells escape programmed cell death [11,12]. One of the desired outcomes of anticancer research is to develop anticancer therapies that can regulate the expression and activity of factors contributing to apoptosis in tumour cells. In one approach, this can be achieved by expressing genes to induce apoptosis or improve tumour chemosensitivity [13]. For instance, by delivering a full-length zymogen form of caspase cDNA or a mutant form of inhibitor of apoptosis protein (IAP), survivin, cell caspase activity can be increased [14].

Gene therapy promises to be a novel strategy for preventing cancer-related deaths. One of the genes that show therapeutic potential is *Death-associated protein kinase 1* (*DAPK-1*). There are hundreds of kinases involved in cell transformation, tumour initiation, proliferation and survival [15,16]. However, the present review focuses on DAPK-1 due to its important roles in cell growth, apoptosis, tumorigenesis, autophagy, inflammation and neurodegeneration. Moreover, DAPK-1 is also the largest member of the DAPK family [17,18]. *DAPK-1* is strictly regulated under physiological conditions, and its deregulation contributes to the development of cancer [18], and its tumour-suppressive role is based on its ability to promote p53-dependent cell death [19,20]. DAPK-1 deregulation also contributes to neurological disorders including ischemic stroke [21] and Alzheimer’s disease [22] by directly phosphorylating Tau on Ser262, resulting in cortical neuron spine damage [23] and also through its involvement in the neurotoxicity of amyloid-β (Aβ) [24,25].

DAPK-1 was discovered as a positive mediator of cell death. Previous findings showed that a reduction in DAPK-1 expression by antisense mRNA prevented γ interferon-induced apoptotic cell death in HeLa cells. Furthermore, HeLa cells died without external stimulus when DAPK expression was elevated [26,27]. The connection between cancer and DAPK-1 was confirmed when the *DAPK-1* promoter regions were shown to be significantly methylated across different types of human tumours in comparison to their corresponding normal tissue samples [28,29,30]. Indeed, the majority of cancers commonly encountered have low expression of DAPK-1 [31,32].

Biological activities in cells are governed by protein–protein interactions, which form the basis of signalling pathways. Several interacting partners of DAPK-1, such as p53, directly and indirectly modify its catalytic activity and pro-apoptotic activity [33,34]. In addition to all known and predicated interactive partners of DAPK-1 shown in Figure 1, there are still many awaiting discovery. For example, it has been reported that DAPK-1 induces autophagy [28], but the interactions between DAPK-1 and autophagy-related proteins have not been elucidated. Thus, there could be some unknown proteins linking DAPK-1 to the downstream signalling pathways. To further understand the functional role of DAPK-1, it is important to identify these unknown DAPK-1 interactive partners.

The *DAPK-1* gene has been reported to be alternatively spliced, producing a wild-type *DAPK-1* and one variant, *s-DAPK-1* [35,36]. Alternative splicing is an important RNA maturation step that allows some exons, or parts of exons and introns, to be kept in mature transcripts, thereby expanding the function and diversity of the proteome. *s-DAPK-1* encodes a polypeptide of 337 amino acids, which contains part of the ankyrin repeat domain, P-loop motif, a portion of the cytoskeletal binding domain of DAPK-1, as well as C-terminal tail extension that is not found in DAPK-1. Analysis of normal human tissue and primary colorectal cancers showed the expression of the *s-DAPK-1* mRNA, indicating that it is expressed in vivo [6].

The function of the s-DAPK-1 isoform remains less clear; however, it has been shown that DAPK-1 possesses both pro-apoptotic and anti-apoptotic functions [37,38]. To elaborate on the role of DAPK-1 in tumour cell survival, a study by Tanaka et al. [38] linked Fas-mediated apoptosis to DAPK-1 expression in human endometrial adenocarcinoma cells. Downregulation of DAPK-1 in HHUA cells significantly enhanced Fas-mediated apoptosis, indicating that endogenous DAPK negatively regulates Fas-mediated apoptosis. It was evident from this study that DAPK-1 can also contribute to cell survival depending on the cellular and environmental context. *DAPK-1* also supports autophagy and apoptosis, thereby suppressing tumour growth and metastasis [19,20].

## 2. Phylogenetics of DAPK-1 Family Members

DAPK-1 was discovered in 1997, and four other kinases with varying degrees of homology to the catalytic domain of DAPK-1 were subsequently identified and reported in humans [19,26,39,40]. These four kinases, which belong to the DAPK family (Figure 2A), include DAPK-2, DAPK-3, DRAK-1 and DRAK-2 [19,26,39,41,42,43,44,45]. As demonstrated in Figure 2B,C, multiple sequence alignment and the phylogenetic tree showed that DRP-1 and ZIP kinases are the closest relatives of DAPK-1 because they share 80% and 83% identity in their catalytic domains compared to DAPK-1, respectively [39]. Similar to DAPK-1, both DAPK-2 and DAPK-3 are regarded as tumour suppressors because they are both implicated in apoptotic cell death [42,43,44,45,46]. The last two family members, the 46 kDa DRAK1 and a 42 kDa DRAK2, have less homology to DAPK-1, with 48% and 51% similarity with the N-terminal kinase domain of the DAPK-1, respectively. These two family members have a less evolutionary relationship with DAPK-1 (Figure 2C), and they lack the CaM regulatory domain and are localized exclusively in the nucleus [46,47,48,49].

The 160-kDa DAPK-1 protein kinase is the largest in the family, consisting of a kinase domain, CaM regulatory domain, eight ankyrin repeats, two phosphate-binding loops (P-loops), Ras of complex proteins C-terminal of ROC (ROC-COR) domain, C-terminal death domain and a 17-amino acid (aa) tail rich in serine residues [26,39,43]. The ankyrin repeats, which are 33-aa motifs in proteins, mediate protein–protein interactions and are particularly important for DAPK-1 degradation. The P-loop, which is an ATP/GTP-binding motif, is widely distributed in proteins [39]. The cytoskeletal-binding domain serves to modulate DAPK-1 catalytic effects and mediates cellular functions through protein–protein interactions [43,44,45]. The C-terminal death domain is a conserved stretch of around 84 aa, which is commonly implicated in protein–protein interactions, kinase activity and apoptosis processes [44].

Lin et al. [36] confirmed that DAPK-1 is alternatively spliced into a smaller variant known as short-DAPK-1 (s-DAPK-1) due to its smaller size in comparison with the full-length DAPK (Figure 3). There is only one report on the expression of s-DAPK-1 in cancer, where the mRNA expression of s-DAPK-1 with that of DAPK-1 in HCT116 colorectal cancer cells, A375 skin cancer cells and non-cancerous kidney cells (Hek-293) were compared [36]. The results of mRNA quantification in cancer cells indicated that, contrary to DAPK-1, s-DAPK-1 is not expressed in colorectal cancer cells, and its mRNA expression in skin cancer cells is significantly higher than that of DAPK-1 [36]. Additionally, it has been found that DAPK-1 is highly expressed in non-cancerous kidney cells compared to s-DAPK-1, understandably so because DAPK-1 serves as a tumour suppressor.

It is important to note that the two standard cellular assays used to determine the function of DAPK-1 include analyzing membrane blebbing and apoptosis effect. Therefore, Lin et al. [36] investigated whether s-DAPK-1 can play a role in these two processes. The results showed that s-DAPK-1 alone does not induce apoptosis upon activation of mitogen-activated extracellular signal-regulated kinase/extracellular signal-regulated kinase (MEK/ERK); however, it can mimic the action of DAPK-1 and cause membrane blebbing. Relative to DAPK-1, s-DAPK-1 has been reported to have a short half-life because of its unique C-terminal tail, which relocates s-DAPK-1 to the cytoplasm and mediates its rapid degradation in a proteasome-dependent manner [35,36]. It has also been shown that transfection of s-DAPK-1 into cells destabilizes DAPK-1; the mechanism is unclear, but it was shown that it is not necessary for s-DAPK-1 to have a polypeptide tail-extension to exert its effect, since its core ankyrin-repeat region is sufficient to mediate the downregulation of DAPK-1 protein by targeting its kinase domain [35,36]. It is possible that s-DAPK-1 targets the kinase domain to mediate DAPK-1 degradation in order to block the ability of this domain to induce membrane blebbing [47].

## 3. Cellular Functions of DAPK-1

In normal tissues, a number of cell types, molecular signals and micro environmental factors work together to maintain tissue function and homeostasis [52]. There are three mechanisms that regulate the homeostasis of cells, and these include proliferation, growth arrest and apoptosis. The multidomain protein kinase DAPK-1 plays a variety of roles, including the regulation of apoptosis, cell survival and autophagy pathways, and each role varies depending on the cellular context and upstream signals [38,53]. We hope to draw more attention to DAPK-1 in these cellular processes by analyzing the recent updates on its biological roles, expression and regulation.

### 3.1. DAPK-1 in Apoptosis

Accidental cell death (ACD) and regulated cell death (RCD) are two types of cell death as shown in Figure 4 [54]. Figure 4 shows the most relevant and noticeable modes of RCD, namely apoptosis, autophagy and necroptosis, which are characterized by a series of typical morphological features. RCD occurs when cells, under the control of various genes, die independently and in a timely manner in order to maintain the stability of the internal environment, whereas ACD refers to a process of cell death that is uncontrolled and initiated by accidental injury stimuli [54,55]. It is believed that programmed cell death, especially apoptosis, ensures that malignant cells do not survive or spread.

Apoptosis is a physiological and pathological process that takes place in an ordered and orchestrated manner. Since it has been established that apoptosis plays an important role in the pathogenesis of many diseases, including cancer, an understanding of its mechanism is vital. One condition where too little apoptosis takes place is cancer, which results in malignant cells that are resistant to induction of cell death [56,57]. There are a number of pathways that are involved in apoptosis, including extrinsic and intrinsic pathways, which can be compromised at any point along the way, resulting in malignant transformation and metastasis of tumour cells and resistance to anticancer therapy [56,57].

The findings that DAPK-1 antisense cDNA reduced IFN-γ and Fas-induced cell death suggest that DAPK-1 may serve as a primary inducer of cell death [19,27]. Furthermore, a great deal of research has been done on the role of DAPK-1 in the Tumour Necrosis Factor α (TNF-α) signalling pathway (Figure 5). Different studies have demonstrated that knocking down DAPK-1 protects DAPK-1 knock-down cells against TNF-α-induced apoptosis, contrary to wild-type [27,58,59]. In addition, some studies with primary cells obtained from mice also showed that DAPK-1 is involved in apoptotic cell death [58]. For example, DAPK-1−/− hippocampal neurons were more resistant to cell death caused by ceramide than their wild counterparts [58], which further supports the notion that DAPK-1 promotes cell death. Additionally, ectopic overexpression of wild-type DAPK-1 has been shown to enhance cell death in various cell lines induced by ceramide and MEK/ERK77 signalling [47,59,60,61]. Wild-type *DAPK-1* has been shown to function as an apoptosis inducer in a variety of cancers including liver cancer [59], cervical cancer [47,61] and breast cancer [47]. DAPK-1 mRNA and protein expression were also shown to be below the limit of detection in some neoplastic-derived B-cell lines, bladder carcinoma, renal cell carcinoma and breast carcinoma cell lines when compared to the positive control cell lines [62].

### 3.2. DAPK-1 in Autophagy

Autophagy is a process by which the body eliminates damaged cells to regenerate new, healthy ones. The process of autophagy unfolds within cells under several stressful conditions, including damage to organelles, expression of abnormal proteins and nutrient deprivation [63]. During autophagy, autophagosomes are formed which capture degraded components and then blend with lysosomes to recycle them [63]. The role of autophagy in cancer biology involves both the promotion and suppression of tumours [64,65].

In addition to apoptosis, DAPK plays a role in the regulation of starvation-induced autophagy in *C. elegans* [66]. DAPK-1 ectopic expression induced caspase-independent cell death in some cell types such as HeLa and MCF-7 cells as demonstrated by the formation of autophagy markers such as phagosomes [47]. These observations suggest that DAPK-1 may be involved in some of the multiple signalling pathways that control autophagy. DAPK-1 is clearly involved in a variety of cell death pathways, but it should be noted that most experiments that implicate DAPK-1 as an important factor in apoptotic cell death were conducted with other upstream signals such as ERK protein or IFN-γ [47,60,61]. The response observed in most cases when DAPK-1 level is increased by ectopic transfection is autophagy, which indicates that DAPK-1 can have dual functions by performing the functions of an autophagy regulator and apoptosis regulator.

### 3.3. DAPK-1 in the Cell Cycle

The progression of the cell cycle is a highly ordered and closely controlled process that involves several checkpoints that monitor signals of extracellular growth, cell size and DNA integrity [67]. Unlike cyclin-dependent kinase inhibitors (CKIs), which act as a brake to stop the progress of the cell cycle in response to regulatory signals, cyclin-dependent kinases (CDKs) and their cyclin partners are the decisive positive regulators of accelerators that promote the progression of the cell cycle. In cancer, cells divide unchecked due to the breakdown in the mechanisms regulating the cell cycle. Cell cycle regulators are commonly altered in human malignancies such as lung cancer [68] and cervical cancer [69], underscoring the significance of maintaining cell cycle commitment in the fight against human cancer.

To understand the role of DAPK-1 in the cell cycle, Wu et al. [67] showed that a cell cycle arrest induced by curcumin in U251 cells falls into G2/M, not G1. Next, they treated DAPK-1 siRNA-transfected cells with curcumin and discovered that the proportion of cells in G2/M phase dropped from 55.2% to 32.9%, thus indicating that DAPK-1 boosts the curcumin-induced cell cycle arrest and apoptosis. Additionally, in order to determine if DAPK-1 was involved in grifolin’s ability to cause G1 arrest, Luo et al. [70] transfected DAPK-1 siRNA into CNE1 cells followed by treatment with grifolin. It was observed that the proportion of cells in G1 dropped from 78% to 48% compared with the siRNA control. These findings allow us to conclude that DAPK-1 is involved in cell cycle regulation, especially in human malignant glioblastoma and nasopharyngeal carcinoma.

### 3.4. Cancer-Related Role of DAPK-1

Tumour-suppressor genes suppress abnormal cell growth and division, in order to sustain the proper balance of our cells. DAPK-1 can be regarded as a tumour suppressor, since its hypermethylation and loss of its expression have been shown in many types of cancers, including cervical cancer [71,72]. Tumour-suppressor gene promoters are often methylated at the CpG-rich regions of their promoters, thus leading to their downregulation and cancer progression [71,73,74]. Several tumour-suppressor genes that are suppressed in this fashion are involved in the regulation of vital biological processes, which include DNA repair, cell cycle and apoptosis [71,72,73]. Some of these tumour suppressors are shown in Table 1; for example, p16 and pRb have their expression lost through their promoter hypermethylation in various cancers, including breast [74,75], cervical [71,72,76] and oral cancers [77].

The loss of DAPK-1 expression in cancer cell lines has been demonstrated to be due to epigenetic silencing via DNA hypermethylation [62]. Numerous malignancies have been linked with *DAPK-1* hypermethylation and loss of its expression, and these include bladder [78], kidney [79], gastric [80], head and neck [81], thyroid [82], lung [31], ovarian [83] and cervical cancers [71,72,76]. Apoptosis is one of the processes that are affected by *DAPK-1* silencing, thus leading to cancer development and progression. This effect was shown when the role of DAPK-1 in topotecan-induced cervical cancer cell death was investigated, and it was discovered that RNA interference-based silencing of DAPK-1 reduced topotecan’s apoptotic effect [84]. In addition, cells without DAPK-1 expression due to promoter hypermethylation were shown to become more invasive and metastatic [85]. It has to be noted that methylation status and expression of the *DAPK-1* gene are not always correlated; for example, a DNA methylation inhibitor, 5′-azadeoxycytidine (5-aza-dC), did not restore DAPK-1 expression for some B cell and lung cancer cell lines [62]. There has also been a loss of DAPK-1 expression in the absence of its promoter hypermethylation [86].
genes-14-01274-t001_Table 1Table 1Various tumour suppressors implicated in human cancers.Tumour SuppressorSilencing MechanismCancer TypeReferencep16Hypermethylation Gastric cancerLung cancerColorectal cancer[87,88,89]pRBHypermethylation,Genomic lossBreast cancerSmall cell lung cancer[74]DAPK-1Hypermethylation Cervical cancerBladder cancerKidney cancerGastric cancerOvarian cancerLung cancer[31,76,78,79,80,83]APCHypermethylation Liver cancerOesophageal cancerColorectal cancerGastric cancerPancreatic cancerHepatic cancer[90]TP53MutationBreast cancerColorectal cancerLung cancer[91,92,93]BRCAMutationBreast cancerOvarian cancerProstate cancer[94,95]MLH1MutationColorectal cancerGastric cancerEndometrial cancer[96]INK4HypermethylationLung cancer[97]RB-1MutationLiver cancerLung cancer[98,99]


## 4. DAPK-1 as a Potential Therapeutic Target

Chemotherapeutic resistance, side effects and non-specific toxicity prompt the search for new anticancer agents. DAPK-1 and its inhibitors have shown potential as therapeutic targets against human diseases [42]. In an Alzheimer’s disease (AD) model, DAPK-1 mediated neuronal cell death. A mouse model of AD has shown that DAPK-1 binds N-myc downstream regulated gene 2 (NDRG2), causing it to be phosphorylated at Ser350 and inducing neuronal cell death. However, DAPK-1 inhibition prevented neuronal cell death triggered by NDRG2 [100]. These data suggest that DAPK-1 could be a novel therapeutic target for treating human Alzheimer’s disease.

Since DAPK-1 is essential for the death and loss of neuronal cells under various stimuli, DAPK-1 inhibitors may offer a novel therapeutic strategy for neurological diseases such as ischemia and Alzheimer’s disease [101]. Alkylated 3-amino-6-phenylpyridazine, the first DAPK-1 inhibitor, was developed to treat sudden brain damage such as stroke [102]. Following a single intraperitoneal injection of alkylated 3-amino-6-phenylpyridazine, it suppressed DAPK-1 kinase activity and decreased brain tissue loss following cerebral ischemia in both acute and sustained brain injury models, suggesting that DAPK-1 inhibition may provide a new therapeutic approach to subdue early programmed cell death in acute brain injury [102].

Pin1 is a crucial enzyme for carcinogenesis because it suppresses tumour suppressors and activates oncogenes [103]. DAPK-1 has been shown to mediate Pin1 phosphorylation, thereby inhibiting its ability to activate transcription factors of oncogenes. As a result of these findings, the tumour suppressor, DAPK-1, might eventually lead to more effective cancer therapeutic strategies by suppressing Pin1 oncogenic activity [43,104].

Combining different treatment strategies for cancer is always preferable, because, in combination, these therapies contribute to overall improved efficacy and safety. A study by Lee et al. [105] investigated the anticancer potential of DAPK-1 and its synergistic combination with Programmed cell death 6 (PDCD6) in ovarian cancer. In order to test whether PDCD6 and DAPK1 have a synergistic or antagonistic effect on apoptosis, Lee et al. [105] measured the proportion of proliferating cells when transfected with PDCD6 cDNA alone, DAPK-1 cDNA alone or both cDNAs co-transfected. Cotransfectants containing both PDCD6 and DAPK1 cDNAs reduced relative proliferation rates compared to transfectants including either PDCD6 cDNA alone or DAPk1 cDNA alone. This suggests that DAPK1 and PDCD6 act in synergy to inhibit cell proliferation.

## 5. Conclusions

In conclusion, *DAPK-1* is arguably one of the most important genes in cell homeostasis and cancer development. The involvement of *DAPK-1* in carcinogenesis is through its epigenetic silencing via hypermethylation of its promoter, a phenomenon that has been found to occur in most human cancers. In this review, we appraised the critical role of DAPK-1 in the regulation of the cell cycle, autophagy and apoptosis and its role in carcinogenesis. Due to its potential as a therapeutic target, a deeper understanding of the role of DAPK-1 in cell death and tumour suppression may result in the development of more effective and specific therapeutic interventions.

## Figures and Tables

**Figure 1 genes-14-01274-f001:**
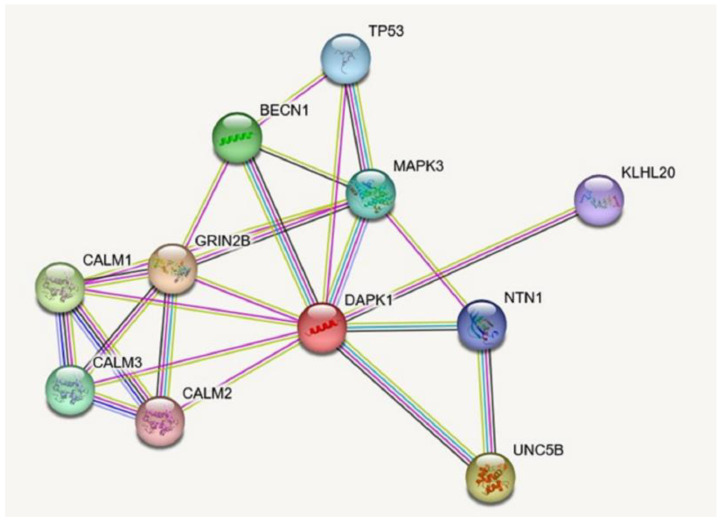
DAPK-1 interacting partners predicted using the STRING online tool.

**Figure 2 genes-14-01274-f002:**
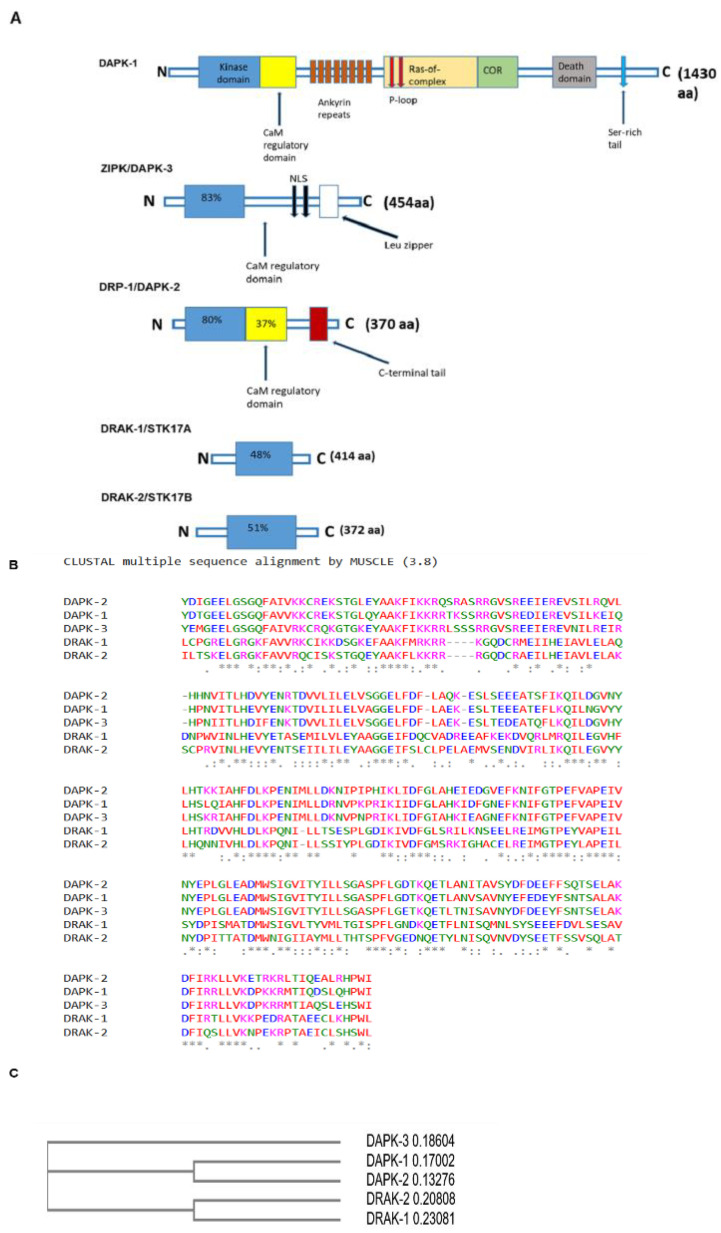
(**A**) DAPK-1 family members and their respective structures showing the kinase domain in all members of the DAPK family. The degree of amino acid identity with DAPK-1 is determined by the number of amino acids within the kinase domains and CaM regulatory domains. Adapted from Farag and Roh, [50], Shiloh et al. [39] and Shoval et al. [51]. (**B**) Sequence alignment of the protein kinase domain found in all members of the DAPK family, the domain was aligned using Multiple sequence alignment tool. The “*” indicates positions where there is one fully conserved residue, while “:” indicates conservation between groups with strongly similar properties (**C**) Phylogenetic tree depicting the relationship between DRAKs and DAPKs in *Homo sapiens*.

**Figure 3 genes-14-01274-f003:**
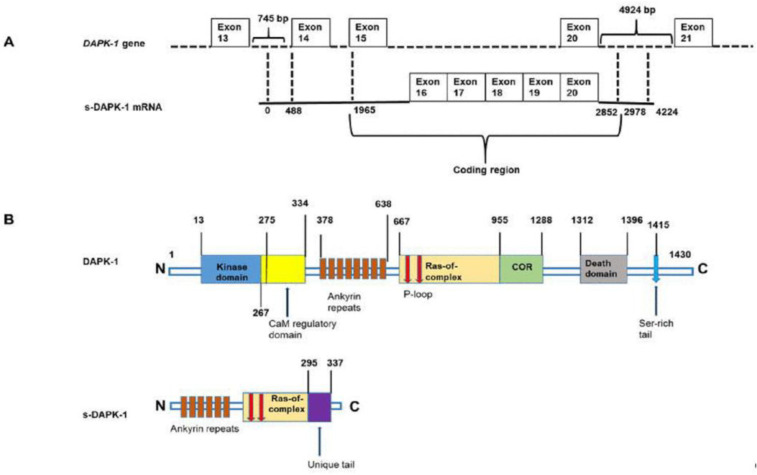
A schematic representation of *s-DAPK-1* mRNA (**A**) and protein (**B**) in relation to DAPK-1. s-DAPK-1′s mRNA begins in intron 13–14 of the DAPK-1 gene, its coding region ends at 126 base pairs in intron 20–21 of DAPK-1. There are 295 amino acids in s-DAPK-1 that are identical to amino acids 447–743 in full-length DAPK-1; however, the last 42 amino acids are unique. Adapted from Lin et al. [36].

**Figure 4 genes-14-01274-f004:**
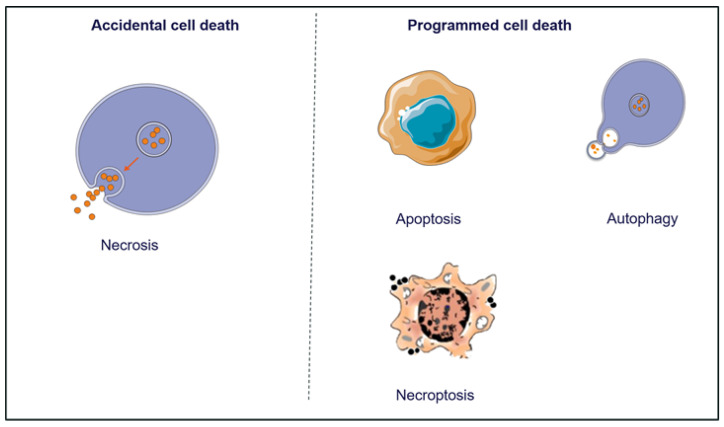
Different types of cell death based on morphological characteristics.

**Figure 5 genes-14-01274-f005:**
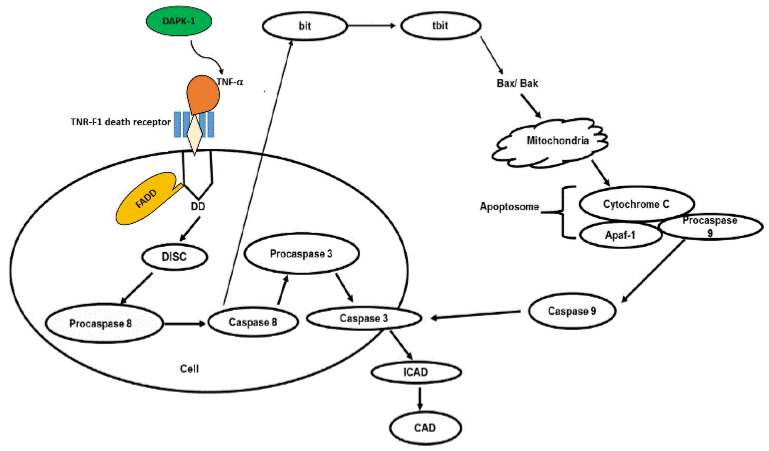
DAPK-1 mediates TNF-α-related apoptosis.

## Data Availability

Not applicable.

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
