# Peer review of "The Role of Death-Associated Protein Kinase-1 in Cell Homeostasis-Related Processes"

_genes, 2023, doi:10.3390/genes14061274_

Round 1
Reviewer 1 Report
Recommendation for Authors
Thank you for your manuscript entitled “Death associated protein kinase-1 mediated cell homeostasis related processes”. I appreciate the effort you put into investigating the role of genetic abnormalities and DAPK-1 in cancer.
The present review clearly summarized the role of DAPK-1 in cell homeostasis-related processes, especially apoptosis, autophagy, and cell cycle as well as the role of DAPK-1 in carcinogenesis.
I believe this article contributes to the existing scientific literature on cancer therapies.
However, I would like to suggest a few revisions before publication. These suggestions aim to enhance the clarity and impact of your work:
Query#1
The current introduction is well written and provides necessary information. However, it is crucial for the authors to better introduce the reason why the authors pay special attention only in DAPK-1, indeed different other kinases play a crucial role in the carcinogenesis of several types of cancer. at this purpose I suggest to the authors to cite the following updated literatures:
- Bhullar KS, Lagarón NO, McGowan EM, et al. Kinase-targeted cancer therapies: progress, challenges and future directions. Mol Cancer. 2018;17(1):48. Published 2018 Feb 19. doi:10.1186/s12943-018-0804-2
- Pecoraro C, Carbone D, Cascioferro SM, Parrino B, Diana P. Multi or Single-Kinase Inhibitors to Counteract Drug Resistance in Cancer: What is New?. Curr Med Chem. 2023;30(7):776-782. doi:10.2174/0929867329666220729152741
Query#2
Section: 2. “Phylogenetics of DAPK-1 family members”. I suggest to the authors to simplify the current section.
Query#3
I kindly ask the authors to revise the title of the article, indeed in the present form, is not mentioned the role of the DAPK-1. I suggest modifying the title including the role of DAPK-1.
The authors are advised to thoroughly revise the style and form of the English language used in the manuscript.
Reviewer 2 Report
This review aims to discuss the current knowledge regarding the mechanisms by which DAPK-1 is involved in cell homeostasis-related processes, specifically apoptosis, autophagy, and the cell cycle. The has also explored how the expression of DAPK-1 impacts carcinogenesis. Considering the implication of DAPK-1 deregulation in cancer development, manipulating DAPK-1 expression or activity could be an interesting promising therapeutic approach for treating cancer. However, the addition of a few comments will be an interest to the readers and a patient outcome.
Major Comments included in the review.
1. Authors should include which cancer type has an advantage when DAPK1 is inhibited.
2. Will combination therapy provide a synergistic effect?
